# Assay for Evaluating the Abundance of *Vibrio cholerae* and Its O1 Serogroup Subpopulation from Water without DNA Extraction

**DOI:** 10.3390/pathogens11030363

**Published:** 2022-03-16

**Authors:** Tania Nasreen, Nora A.S. Hussain, Jia Yee Ho, Vanessa Zhi Jie Aw, Munirul Alam, Stephanie K. Yanow, Yann F. Boucher

**Affiliations:** 1Department of Biological Sciences, University of Alberta, Edmonton, AB T6G 2E9, Canada; tnasreen@ualberta.ca (T.N.); nahussai@ualberta.ca (N.A.S.H.); 2Singapore Centre for Environmental Life Sciences Engineering (SCELSE), National University of Singapore, Singapore 637551, Singapore; jiayeeho@nus.edu.sg (J.Y.H.); vanessa.azj@nus.edu.sg (V.Z.J.A.); 3Saw Swee Hock School of Public Health, National University Health System, National University of Singapore, Singapore 117549, Singapore; 4Centre for Communicable Diseases, International Centre for Diarrhoeal Disease Research, Bangladesh (ICDDR, B), Dhaka 1212, Bangladesh; munirul@icddrb.org; 5School of Public Health, University of Alberta, Edmonton, AB T6G 2E9, Canada; yanow@ualberta.ca; 6Department of Medical Microbiology and Immunology, University of Alberta, Edmonton, AB T6G 2E9, Canada

**Keywords:** cholera, *Vibrio cholerae*, *Vibrio cholerae* O1, endemic, toxigenic, abundance, qPCR

## Abstract

Cholera is a severe diarrheal disease caused by *Vibrio cholerae*, a natural inhabitant of brackish water. Effective control of cholera outbreaks depends on prompt detection of the pathogen from clinical specimens and tracking its source in the environment. Although the epidemiology of cholera is well studied, rapid detection of *V. cholerae* remains a challenge, and data on its abundance in environmental sources are limited. Here, we describe a sensitive molecular quantification assay by qPCR, which can be used on-site in low-resource settings on water without the need for DNA extraction. This newly optimized method exhibited 100% specificity for total *V. cholerae* as well as *V. cholerae* O1 and allowed detection of as few as three target CFU per reaction. The limit of detection is as low as 5 × 10^3^ CFU/L of water after concentrating biomass from the sample. The ability to perform qPCR on water samples without DNA extraction, portable features of the equipment, stability of the reagents at 4 °C and user-friendly online software facilitate fast quantitative analysis of *V. cholerae*. These characteristics make this assay extremely useful for field research in resource-poor settings and could support continuous monitoring in cholera-endemic areas.

## 1. Introduction

Cholera is a life-threatening diarrheal disease caused by pathogenic strains of *Vibrio cholerae*. Today, cholera still perseveres as a global threat to public health due to its high morbidity and mortality rates [1,2,3,4,5]. There were around 250,000 suspected cholera cases and over 3500 deaths reported in 2021 [6], and an estimated 2.9 million cases and 95,000 deaths occur each year around the world [7]. The discrepancy of cholera cases in 2021 could be due to the public health measures and nonpharmaceutical interventions reinforced due to COVID-19 pandemic globally [8,9,10]. Lack of access to safe drinking water and inadequate management of sanitary systems in resource-poor countries are two major reasons this disease remains a significant public health problem. Current data on the global disease burden of cholera identify 47 countries around the world affected by cholera, and over half a billion people reside in areas that are labelled as cholera hotspots [11]. Cholera outbreaks in recent years affecting Haiti, Somalia, Ethiopia, Iraq, Mozambique, Zambia, Sudan, Nepal and Zimbabwe demonstrated that devastation—from an earthquake, for example—can cause an outbreak [12]. In 2018, the epidemic in Yemen was reported to be the world’s fastest-growing outbreak, where 10,000 suspected cases were reported on a weekly basis [13]. The cumulative impact of cholera in Yemen currently sits at a death toll of nearly four thousand since October 2016, with over 2.5 million people infected so far [14].

*V. cholerae* is naturally found worldwide, especially in brackish riverine, coastal and estuarine environments [15]. Not all *V. cholerae* present in nature are pathogenic, and only a subset of strains is a known threat to humans. Amongst the 200 serogroups of this species, only O1 and O139 are associated with cholera cases (toxigenic serogroups) and are responsible for epidemic and pandemic cholera outbreaks [16,17,18].

Estimation of *V. cholerae* abundance, along with that of its pandemic generating serogroups [19] in aquatic ecosystems, is difficult because of the high spatio-temporal variability exhibited by its natural populations [20]. Epidemiological studies and analysis of cholera outbreaks revealed that the disease occurs in a regular seasonal pattern in cholera-endemic areas [21,22,23] and causes outbreaks only under certain conditions, which may be attributed to environmental and climatic factors—for example, heavy rainfall followed by blooms of phytoplankton and zooplankton [24,25,26,27].

Although cholera has been endemic to the Ganges Delta for centuries, it is an imported disease in most other locales, where it can vanish after a single outbreak or linger for decades before disappearing. For example, Haiti had no recorded cholera for centuries [28,29,30,31], but the disease rapidly spread after the 2010 earthquake, which devastated infrastructure in the country, and with the arrival of United Nations (UN) troops carrying the bacteria from Nepal [31,32]. The country then faced a decade of the cholera epidemic before the disease faded in 2019–2020 [33]. Because of these epidemiological dynamics, cholera has been categorized as an emerging and re-emerging infectious disease [34,35].

In addition, surveillance of cholera outbreaks through clinical diagnosis provides an estimation of the associated disease burden, but it is unable to provide quantitative information of the pathogen or its abundance in source environments. To achieve a better understanding of the prevalence of cholera and to direct appropriate control measures and treatment, it is necessary to identify and promptly quantify its causative agent in its reservoir. However, one of the major obstacles is that the number of *V. cholerae* (toxigenic and non-toxigenic) is often below the limit of detection of current field analytical methods, even during outbreaks [36].

Conventional culture methods remain the gold standard for laboratory diagnosis of cholera [37,38] and often require pre-enrichment of the sample [39]. Moreover, two to three days are required to conduct testing, even with modern laboratory infrastructure. With these methods, isolation and identification are possible, but the total abundance quantified is often underestimated, as a considerable proportion of most populations exists as viable but non-culturable cells (VBNC), which do not revive upon culturing on microbiological media [39,40]. In areas with limited or no laboratory facilities, simple dark-field microscopy is used to detect characteristic movement of *V. cholerae* in stool samples, but this method is not feasible on water sources where the bacterium is much more dilute [41]. The Crystal^®^ VC Rapid Diagnostic Test (RDT) is also used for point-of-care detection to predict potential cholera outbreaks. This diagnostic test uses stool samples and is mainly based on detection of either cholera toxin [42,43] or a lipopolysaccharide antigen [42,44,45]. It is useful to detect serogroups O1 and O139. However, due to the poor sensitivity and specificity of this method, additional tests are required to confirm the presence of toxigenic *V. cholerae* in stool samples [42]. A Direct Fluorescent Antibody (DFA) technique using polyclonal anti-O1 serum is also used to detect *V. cholerae* O1 in smears prepared from samples. This procedure has been used for both clinical and pre-enriched environmental samples [39,46]. Catalyzed reporter deposition fluorescence in situ hybridization (CARD-FISH) in combination with solid-phase cytometry is also a new protocol for rapid, specific, and sensitive cell-based quantification [47,48]. Despite these methods facilitating cholera diagnosis, most of them are qualitative, only detect *V. cholerae* O1 or O139 serogroup and have been optimized for clinical but not environmental specimens.

To improve cholera surveillance, it is essential to accurately determine the abundance of the total and disease-causing serogroups of *V. cholerae* in its reservoir. To achieve this goal, we designed two species-specific primers and probes to precisely detect and quantify total *V. cholerae* and *V. cholerae* O1 serotype from environmental water samples without DNA extraction. Employing the Chai Open qPCR equipment—a low cost, portable qPCR machine—we demonstrate the potential for our designed set of qPCR primers to be utilized during field sampling. In-field quantification of prokaryotic and eukaryotic constituents of interest using Open qPCR have been recently performed on recreational waters in Alberta and present comparable results to conventional qPCR equipment [49]. Altogether, this method can determine the absolute abundance of this bacterium and the main serogroup associated with pathogenicity across hundreds of samples in a spatio-temporal gradient, making it possible to pinpoint the source of cholera outbreaks and warn of potential outbreaks.

## 2. Results

### 2.1. Analytical Validation

The qPCR assay was validated by using a blind panel of filter-sterilized environmental water samples collected from the Gabtoli area (Dhaka, Bangladesh) (Figure 1) and spiked with *V. cholerae* reference strains, other vibrio species and non-vibrio species.

Forty-six bacterial isolates of known concentration of 3 × 10^4^ CFU/mL (CFU—colony forming unit) were tested, including non-O1 *V. cholerae*, *V. cholerae* O1, other *Vibrio* species (*V. parahaemolyticus*, *V. vulnificus*, *V. metoecus*, *V. mimicus*), as well as non-*Vibrio* species (*Escherichia coli* and *Pseudomonas aeruginosa*) (Table 1). All seventeen non-O1 *V. cholerae* and eight *V. cholerae* O1 were positive for the *viuB* gene. Only samples spiked with *V. cholerae* O1 strains were positive for the *rfbO1* gene. The other *Vibrio* spp. and non-*Vibrio* species were negative for these two gene targets. Thus, the analytical specificity (i.e., the ability of an assay to detect and/or measure a specific organism in a sample) of this method was 100% (Table 1). Analytical sensitivity was also found to be 100% based on detection of 3 CFU/reaction (Table 2).

### 2.2. Limit of Detection (LOD)

The LOD of the assay was determined as 3 CFU per reaction from the standard curve constructed using serially diluted standards of the *V. cholerae* El Tor O1 N16961 reference strain (Figure 2). We also determined a Sample Limit of Detection (SLOD) of 5 × 10^3^ CFU/L for the filter-sterilized environmental water samples (5 mL) spiked with a known number of *V. cholerae* N16961 concentrated to ~10 μL with an Amicon ultra-0.5 centrifugal filter device (Appendix A).

### 2.3. Assay Precision and Efficiency

Following the Minimum Information for Publication of Quantitative qPCR Experiments (MIQE) guidelines, intra-assay variation (variation between replicates in the same experiment) and inter-assay variation (variation between replicates from different experiments) were evaluated to determine the repeatability and reproducibility of the assay for detecting and quantifying total *V. cholerae* and its O1 serogroup subpopulation. Precision analysis to test random variation of repeated measurements was done for this assay by calculating the coefficient of variation (%CV) of multiple replicates of standards run in the same experiment and experiments on different days. Intra-assay %CV ranged from 0.01 to 0.05% for the *viuB* assay and 0.01 to 0.03% for the *rfbO1* assay. Inter-assay %CV ranged from 0.07 to 0.20% for *viuB* and 0.03 to 0.21% for *rfbO1* (Table 2). The efficiency of both assays was 100% based on the standard curve generated from a serial dilution of *V. cholerae* N16961 (Figure 2) with R^2^ = 0.99 and slope of −3.3.

To test for qPCR inhibition, we compared the quantitative cycle value (Cq) values for different dilutions of filter-sterilized environmental water samples spiked with the reference *V. cholerae* strains. The 10× diluted samples shifted the Cq values by 3.3 ± 0.07 cycles (Appendix A), indicating no significant inhibition.

### 2.4. Analysis of Environmental Water Samples

With this field-ready method, it was possible to quantify the abundance of both total *V. cholerae* and *V. cholerae* O1 in the same experiment (Figure 3). Five mL of environmental water samples were concentrated to 100 μL, from which 10 μL was used in the qPCR amplification. Each reaction was run in triplicate. The abundance of *V. cholerae* in the water samples collected from the Gabtoli area, Dhaka, Bangladesh was between 3.7 × 10^4^ to 3.9 × 10^4^ CFU/L (Figure 4). *V. cholerae* O1 was found at 4.7 × 10^3^ to 5.4 × 10^3^ CFU/L, representing approximately 13% of the total *V. cholerae* population (Figure 4).

## 3. Discussion

Since cholera is a waterborne infectious disease and the primary mode of transmission is via the fecal-oral route, environmental water bodies serve as an inevitable reservoir for pathogenic *V. cholerae*. This bacterium is associated with plankton mainly in brackish waters and is ubiquitous in temperate and tropical estuarine microbial communities [65]. As a survival strategy, *V. cholerae* goes into a viable but non-culturable state (VBNC) under unfavorable environmental conditions, in which it assumes a coccoid shape and cannot be cultured using traditional culturing methods [24,66]. Non-culturable *V. cholerae* in biofilm were reported in environmental water samples from Mathbaria (Bangladesh), and were able to resume active growth after passage through the gastrointestinal tract of rabbits (in animal passage of non-culturable *V. cholerae* O1) following a period of more than a year in a microcosm [67]. Therefore, passing through the human host could be a means of revival from the VBNC state, contributing to the amplification of *V. cholerae* prior to an outbreak [68,69] and its subsequent transmission through the fecal-oral route due to poor management of drinking water and hygiene. This survival state frequently found in the environment for *V. cholerae* means that water bodies can serve as a long-term reservoir for this pathogen, leading to the consistent and persistent pattern of cholera epidemics historically documented on the coast of Bangladesh. There is also recent evidence that indicates that environmental *V. cholerae* may play a role in reviving VBNCs through the over- production of products associated with quorum sensing [38]. As the body of literature regarding environmental *V. cholerae* increases [15,20,27,70], we expect the interactions between pathogenic and non-pathogenic *V. cholerae* to be resolved in greater detail.

The ability of *V. cholerae* to exist in a VBNC state has hindered our ability to quantify it in environmental reservoirs using traditional methods. Limited resources and lack of infrastructure in countries where cholera is endemic have also increased the challenge in monitoring its causative agent. The qPCR method developed here allows the detection and quantification of *V. cholerae* and the O1 serogroup strains responsible for most outbreaks. Furthermore, the portable and low-cost instrument used (Chai Open qPCR), as well as our streamlined protocol, allow processing of water samples on site, without the need for DNA extraction or any pre-enrichment procedure. The main challenge for direct quantification is that the number of bacteria in environmental water is usually below the LOD of existing qPCR methods [71]. However, this problem was overcome by increasing the concentration of the water sample 50-fold using simple size-exclusion centrifugation.

The targeted *viuB* gene sequence is only present in *V. cholerae* and can be amplified from both O1 and non-O1 strains. Moreover, *V. cholerae* O1 could be detected with primers targeting the *rfbO1* gene, thus allowing the determination of the abundance of both populations with 100% specificity. The assay sensitivity was as low as three CFU per reaction, which is expected for a well-designed and sensitive assay, as described in the MIQE guidelines [72]. The limit of detection for concentrated water samples was 5 × 10^3^ CFU/L, which is at the lower limit of an infectious dose of *V. cholerae*, reported to be 10^3^–10^9^ cells based on a range of factors, such as the health of the exposed individual [73]. More specifically, the infectious dose for toxigenic *V. cholerae* O1 is typically around 10^4^–10^6^ cells, whereas the infective dose for non-O1 strains is around 10^6^–10^9^ [74]. Without the concentration procedure, the limit of detection of *V. cholerae* in environmental water samples would be 3 × 10^6^ CFU/L, which is still useful to determine if a strong risk of contracting cholera exists. This assay determines the absolute abundance of both *V. cholerae* O1 and non-O1 in parallel qPCR reactions and thus is very useful to calculate the proportion of toxigenic *V. cholerae* O1 in a particular geographical location. Therefore, this assay is useful to track cholera in endemic areas like Bangladesh, where all detected *V. cholerae* O1 strains were found to be toxigenic [67,75].

The possibility of storing the qPCR master mix on ice (4 °C) for extended periods of time and the portable feature of the Chai thermocycler (small footprint of 28 cm × 24 cm, weight of 4 kg) facilitate the use of this method in field-level surveillance and also avoid transportation problems leading to deterioration of the samples. The convenient setup of the whole procedure makes this assay workable in most resource-limited settings. Moreover, quick processing of the sample reduces the chance of cross-contamination.

As proof of the concept, we analyzed water samples collected from the river basin of the Turag River in a region endemic for cholera, the Gabtoli area of Dhaka, Bangladesh. Approximately 25,000 people live in this area, which is surrounded by brickfields with high traffic [76]. This site was chosen because previous studies indicated the presence of *V. cholerae* O1 and cholera infections in people living in the surrounding area [77]. The number of total *V. cholerae* detected was about 3.9 × 10^4^ CFU/L, 13% of which was *V. cholerae* O1 (~5 × 10^3^ CFU/L). This is of concern, as it is within the range of the infectious dose for cholera if someone were to ingest a few hundred milliliters of water (>10^3^ cells) [73]. Previous studies based on the Crystal VC^®^ dipstick test after sample enrichment for 18 h in alkaline peptone water showed that in urban Dhaka, Bangladesh, 30% of water sources used by households of cholera patients were contaminated with *V. cholerae* [78,79]. From our observations during sampling, which are also reported by other studies, the water bodies around Dhaka city serve as drinking water sources and are frequently used for domestic purposes, such as washing utensils and bathing [80]. It is therefore likely that local rivers and ponds serve as important reservoirs of the cholera pathogen in Dhaka city. The importance of local ecosystems in maintaining toxigenic cholera is known [81,82], but because of the difficulty in monitoring at sufficient frequency in a number of locations, a direct link has never been established between a rise in *V. cholerae* numbers and the start of a seasonal epidemic, despite the timing of these being well known from the tracking of cholerae cases in hospitals.

Many qPCR assays for the quantification of *V. cholerae* have been previously developed, but all of them require modern laboratory facilities and are not amenable to extensive field studies. Furthermore, there is not a single study of the environmental tracking of *V. cholerae* in a cholera-endemic setting of significant size. With the currently available techniques, either spiked environmental samples were evaluated after pre-enrichment in alkaline peptone water or DNA extraction after filtration [16,77,83]. Amongst these methods, culture-based techniques are still the most frequently used and only reveal a small proportion of naturally occurring bacterial populations in water samples. These methods do not capture coccoid non-culturable cells within clusters of biofilms in estuarine environments present in Bangladesh water bodies during the period between outbreaks when reported cases of cholera are low [84].

Another widely used method is dipstick, which is valuable to detect *V. cholerae* O1 and O139 in both water and stool samples. This is a rapid and inexpensive method but has a limit of detection of 10^7^ CFU of enriched culture, which is higher than the usual infectious dose of cholera [73]. This method only provides a qualitative output and requires six hours of enrichment before testing. Also, because of compromised sensitivity, it was recommended for use in combination with a traditional bacterial culture method [78].

Our assay is specific, sensitive, and convenient for field studies, making it possible to overcome the limitations of current “rapid” techniques. In this study, a single channel Chai thermocycler was used, which only permitted FAM to be used as a fluorescein dye. A dual channel Chai system also exists, where HEX or VIC fluorescein dyes can also be used with FAM, making a multiplex qPCR assay possible. Thus, it is possible to optimize a multiplex assay where *V. cholerae* O1 can be rapidly identified and quantified simultaneously in a single reaction along with total *V. cholerae*.

During the environmental sample analysis, we did not find significant inhibition to PCR amplification in our assay. However, for testing water from more contaminated sources, as determined by multivariate analysis of environmental factors where the industrial waste disposal and high population density matters [85,86], further treatment of the sample to remove inhibitors may be required. A study on water quality assessment of the roadside surface of Savar, Dhaka explained the impact of vehicle emission, atmospheric deposition from brick fields, industrial pollution and massive urbanization on the water reservoirs; total suspended solids (>25 mg/L), total dissolved solids (>840 mg/L), biological oxygen demands (0.758 mg/L) and dissolved oxygen (4.5 mg/L) were high or even higher than the standards [76]. Another potential limitation of our assay would be a discrepancy in genome copy numbers per cell between the standard and the samples. This potential source of error is minimized by using filter-sterilized environmental water from the same location as the samples for the standards. Live cells of a known concentration are inoculated and incubated in this water for 30 min prior to performing qPCR to adjust their physiological state.

In conclusion, we developed a species-specific qPCR method that can be conveniently used at the sampling site without any need for pre-enrichment or DNA extraction. This test can support a continuous monitoring program for *V. cholerae* O1 in water reservoirs used by residents, which in coordination with local authorities could limit risks of contracting cholera and allow the identification of sources of contamination.

## 4. Materials and Methods

### 4.1. Preparation of Bacterial Cultures

Forty-six different strains of *Vibrio* spp. and other gammaproteobacteria were used to validate analytical sensitivity and specificity of the assay in this study (Table 1). Seventeen non-O1 *V. cholerae* from environmental sources, eight *V. cholerae* O1 from both clinical and environmental sources, and ten *V. metoecus* from environmental sources were tested (Table 1). Multiple strains from three other *Vibrio* species [*V. parahaemolyticus* (1), *V. vulnificus* (2) and *V. mimicus* (3)] were tested along with two strains of *Pseudomonas aeruginosa* and *Escherichia coli* each. The *Vibrio* strains were grown in tryptic soy broth (TSB) (Becton Dickinson, Franklin Lakes, NJ, USA) with 1.0% NaCl (BDH) at 30 °C and 200 rpm overnight in a shaking incubator.

### 4.2. Collection and Processing of Environmental Samples

Environmental water samples were collected from the river basin of the Turag river in the Gabtoli area (the coordinates of the latitude and longitude are 23.783726, 90.344246) [87] of Dhaka, Bangladesh (Figure 1) during August 2018. Three locations, approximately 5 m apart from each other, were chosen at this site and water samples were collected in triplicate from each of those locations (nine samples in total). Five milliliters of surface water was collected directly in Thermo Scientific Nunc 50 mL conical centrifuge tubes (Thermo Fisher Scientific, Seoul, Korea). When needed, collected water samples were concentrated using an Amicon Ultra-0.5 centrifugal filter device (Merk KGaA, Darmstadt, Germany) (Figure 5) according to the user’s manual. Ten Amicon tubes, each containing 500 μL of water sample, were centrifuged at 14,000× *g* for 20 min in a mini spin plus centrifuge (Eppendorf, Hamburg, Germany). All the concentrates were pooled. Altogether, 5 mL of water from each of the three replicates from each of three locations were concentrated to 100 μL.

### 4.3. Amplification Using the Chai Open qPCR Platform

Previously developed species-specific primers [77] were used to detect and quantify *V. cholerae* in this assay. Briefly, the *viuB* gene encoding vibriobactin utilization protein B was used to quantify total *V. cholerae* (all serogroups), as it is a single-copy gene present in all *V. cholerae* with only divergent homologs present in other species. To detect and quantify *V. cholerae* from the O1 serogroup, we used specific primers and probes to amplify the *rfbO1* gene essential for the synthesis of this antigen (Table 3) [77]. For this study, the reporter dye 56-FAM (Fluorescein) was used for both probe sets, as this qPCR assay was optimized in the Chai Open qPCR thermocycler (CHAI, Santa Clara, CA, SA), which has a single channel to detect wavelengths of 513–555 nm.

Dynamite qPCR Master mix used in this study is a proprietary mix, developed and distributed by the Molecular Biology Service Unit (University of Alberta, Edmonton, AB, Canada). It contains Tris (pH 8.3), KCl, MgCl_2_, glycerol, Tween 20, DMSO, dNTPs, ROX as a normalizing dye and antibody-inhibited Taq polymerase. The volume of each PCR reaction was 50 μL, which contained 25 μL of 2× Dynamite qPCR master mix, 5 μL of 500 nM primer-250 nM probe mix, 10 μL of molecular grade water and 10 μL of the concentrated environmental water sample. The recommended long-term storage temperature for this master mix is 4 °C. Conditions for the real-time qPCR are as follows: initial activation of the enzyme at 95 °C for 2 min followed by 45 cycles of 95 °C for 30 s, 60 °C for 1 min in the Chai Open qPCR system. In this qPCR method, the detection and quantification of the *viuB* and *rfbO1* gene markers were carried out in parallel in separate tubes but in the same thermocycler run. Standards of a known number of cells and no template control were included in every assay.

### 4.4. Standard Curve for the qPCR Assay

Standard curves were generated using the *V. cholerae El Tor* O1 N16961 reference strain. Pure bacterial culture was grown on LB agar (Becton Dickinson) at 30 °C overnight. Bacteria were then diluted in sterile water prepared by filtering 50 mL of water from the study location pushed through a 0.2 μm PES filter media (Whatman, GE Healthcare, Amersham, UK) using a 50 mL syringe (BD, USA). A series of standards were prepared in which bacterial cells were added at 3 × 10^5^ CFU, 3 × 10^4^ CFU, 3 × 10^3^ CFU, 3 × 10^2^ CFU, 30 CFU and 3 CFU per 10 μL of filter-sterilized water. Inoculum concentrations were quantified using a standard drop plate method [88]. Standards were incubated for 30 min at room temperature before adding an inoculum to the qPCR reaction so that the cells could acclimatize to their new environmental conditions. Standards were run in three independent experiments, with three replicates per dilution and repeated on three different days. The average of each experiment was assessed to define intra- and inter-assay variation (Table 2). A standard curve was generated by plotting the log value of the calculated CFU per reaction against the Cq. The Cq value is the cycle at which the fluorescence from amplification exceeded the background fluorescence in the MIQE guidelines [72]. The unit CFU/L used throughout the manuscript indicates CFU equivalent/L, as not all cells present in the samples or standards would actually grow on solid media after being exposed to room-temperature freshwater.

### 4.5. Determination of the LOD and qPCR Efficiency

The LOD of the assay was determined from the standard curve constructed from serially diluted standards of *V. cholerae* N16961, as mentioned above (Figure 2). To determine the LOD of samples that were concentrated, known numbers of *V. cholerae* N16961 were spiked in filter-sterilized environmental water samples collected from the same location (Gabtoli, Dhaka, Bangladesh) and subjected to the same concentration protocol as environmental samples using the Amicon ultra-0.5 centrifugal filter device. 1 mL of spiked environmental water sample was concentrated to 20 μL after 30 min of spinning. The qPCR assay was carried out on these concentrated samples containing known bacterial cell numbers in the Chai Open qPCR system (Appendix A). Cq values were used to define the LOD of the assay. The LOD typically is assumed to be the highest Cq value observed for the lowest concentration that can be determined based on the dilution at which all replicates were positive across ten repeated experiments.

The qPCR efficiency of the assay was calculated in Excel using the following formula: Efficiency = 10^[−1/Slope]^ [16,89].

## Figures and Tables

**Figure 1 pathogens-11-00363-f001:**
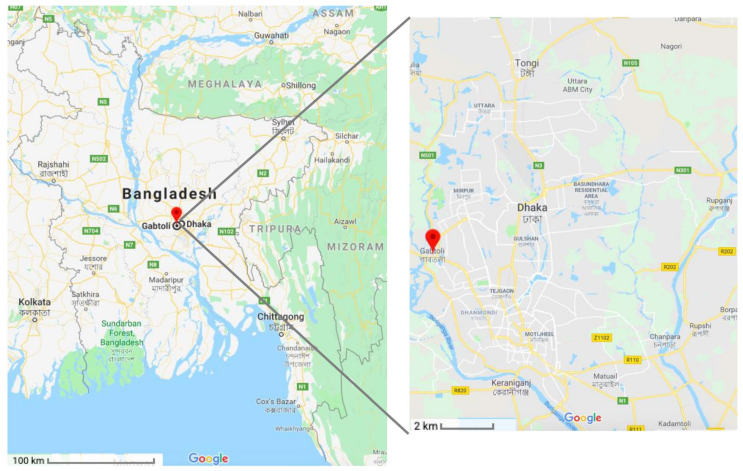
Environmental water sampling sites in Bangladesh. An endemic site for *V. cholerae* is shown on the map of Bangladesh. Environmental water samples were collected from the river basin of the Turag river in the Gabtoli area in Dhaka city, an inland region of Bangladesh. Samples were collected in triplicate from three locations that were 5 m apart (Image source: Google).

**Figure 2 pathogens-11-00363-f002:**
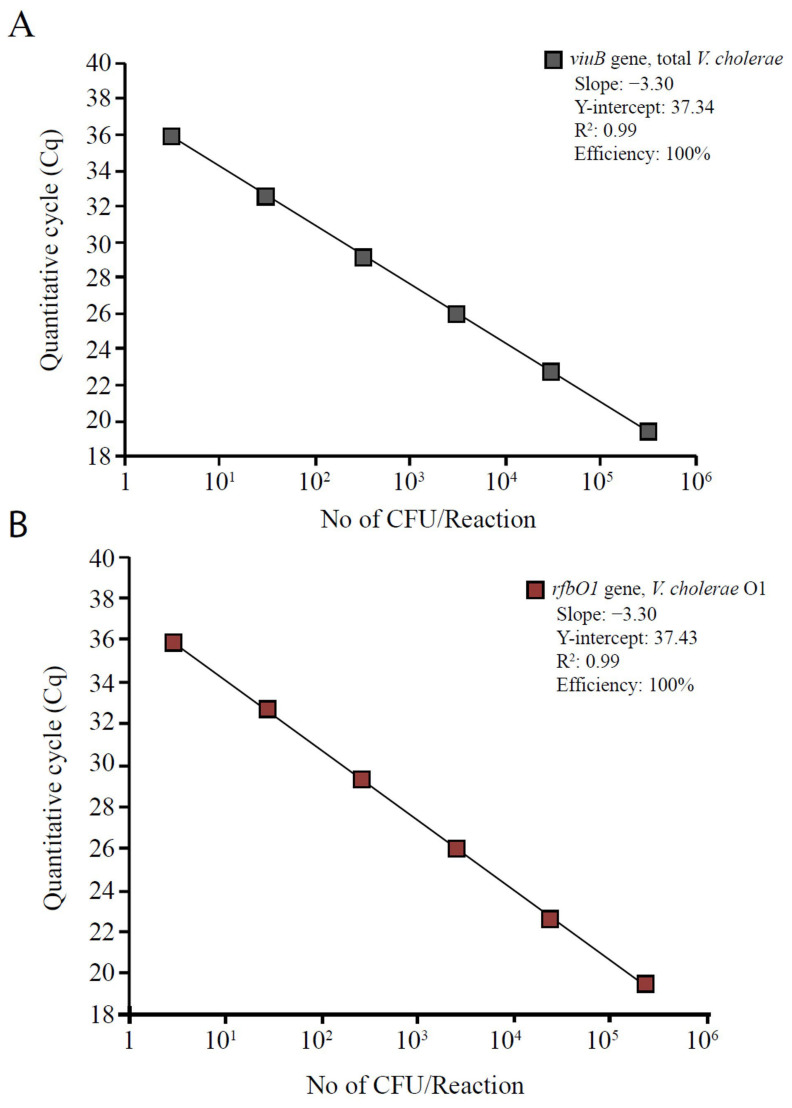
Standard curves for detection and quantification of total *V. cholerae* and *V. cholerae* O1 by qPCR. Two gene markers with fluorogenic probes, (**A**) *viuB* (*V. cholerae* specific) and (**B**) *rfbO1* (*V. cholerae* O1 specific) were used. Cells of reference culture (*V. cholerae* N16961 El Tor O1) were serially diluted 10-fold to yield concentrations ranging 3 to 3 × 10^6^ CFU per reaction (from left to right). Fluorescence was measured in relative units. Each reaction was done in triplicate.

**Figure 3 pathogens-11-00363-f003:**
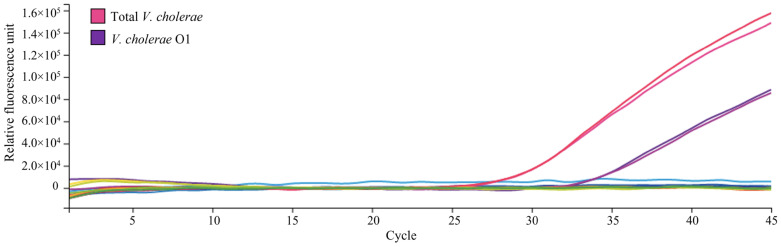
Detection and quantification of total *V. cholerae* and *V. cholerae* O1 in the same qPCR experiment. Filter-sterilized environmental water samples collected from the Gabtoli site were inoculated with known concentrations of cells of reference strain *V. cholerae* N16961 El Tor O1. Quantification was performed with both *viuB* and *rfbO1* primers in the same PCR run to detect total *V. cholerae* and O1 serogroup *V. cholerae*, respectively. Hot pink and red curves indicate total *V. cholerae*, purple and violet curves indicate *V. cholerae* O1, and all other curves indicate negative controls.

**Figure 4 pathogens-11-00363-f004:**
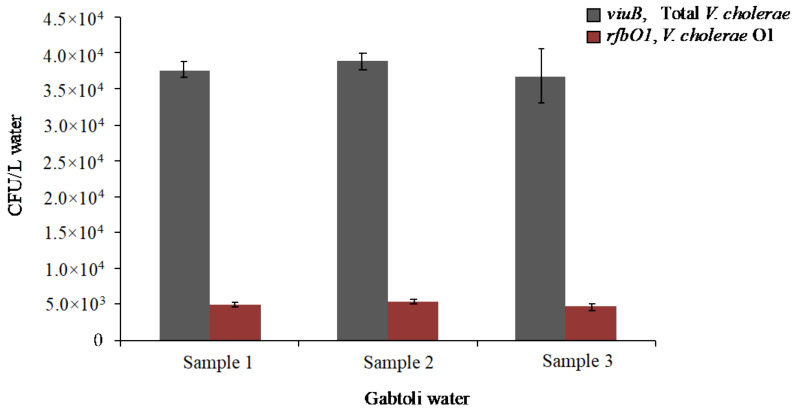
Abundance of *V. cholerae* along with its O1 serogroup subpopulation in water from an inland urban region (Gabtoli, Dhaka) of Bangladesh. Water samples were collected in August 2018 and analyzed using the developed qPCR assay. The *viuB* gene marker was used to quantify total *V. cholerae*, and *rfbO1* was used to quantify *V. cholerae* O1. Each qPCR reaction was run in triplicate and evaluated with corresponding standards. Error bars represent the standard deviation of means from biological replicates.

**Figure 5 pathogens-11-00363-f005:**
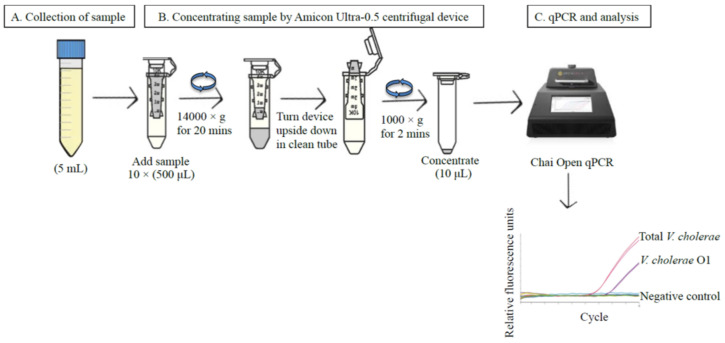
Method for processing of environmental samples for qPCR. (**A**) Sample collection. (**B**) Concentrating sample by Amicon Ultra-0.5 centrifugal device. (**C**) qPCR in the Chai Open qPCR thermocycler (https://www.chaibio.com/openqpcr (accessed on 5 February 2022). 5 mL of water was collected from each location. Ten Amicon tubes, each with 500 μL of water sample, were centrifuged at 14,000× *g* for 20 min in a mini spin plus centrifuge, and all the concentrates were pooled in a total volume of concentrate of 100 μL.

**Table 1 pathogens-11-00363-t001:** Bacterial strains used in the analytical validation of this assay. “+” indicates presence of the mentioned target gene, while “−” indicates absence.

Species	No. of Strains	Strain	Target Genes		Source	Reference
			*viuB*	*rfbO1*		
*V. cholerae*						
*V. cholerae* non-O1	17	OYP1G01	+	−	Environmental	This study
		OYP2A12	+	−	Environmental	This study
		OYP2E01	+	−	Environmental	This Study
		OYP3B05	+	−	Environmental	[50]
		OYP3F10	+	−	Environmental	This study
		OYP4B01	+	−	Environmental	This study
		OYP4C07	+	−	Environmental	[50]
		OYP4G08	+	−	Environmental	This study
		OYP4H06	+	−	Environmental	This study
		OYP4H11	+	−	Environmental	This study
		OYP6D06	+	−	Environmental	This study
		OYP6E07	+	−	Environmental	This study
		OYP6F08	+	−	Environmental	This study
		OYP6F10	+	−	Environmental	This study
		OYP7C09	+	−	Environmental	This study
		OYP8C06	+	−	Environmental	This study
		OYP8F12	+	−	Environmental	This study
*V. cholerae* O1	8	N16961	+	+	Clinical	[51]
		V52	+	+	Clinical	[52]
		EDC-728	+	+	Environmental	This study
		EDC-753	+	+	Environmental	This study
		EDC-754	+	+	Environmental	This study
		EDC-755	+	+	Environmental	This study
		EDC-772	+	+	Environmental	This study
		EDC-805	+	+	Environmental	This study
Other *Vibrio* species						
*V. parahaemolyticus*	1	ATCC 17802	−	−	Clinical	[53]
*V. vulnificus*	3	ATCC 27562	−	−	Clinical	[54]
		MO6-24	−	−	Clinical	[55]
		CECT 5769	−	−	Environmental	[56]
*V. metoecus*	10	RC341	−	−	Environmental	[57]
		OP3H	−	−	Environmental	[58]
		OYP4D01	−	−	Environmental	[59]
		OYP4E03	−	−	Environmental	This study
		OYP5B04	−	−	Environmental	[59]
		OYP5B06	−	−	Environmental	[59]
		OYP5H08	−	−	Environmental	This study
		OYP8G05	−	−	Environmental	This study
		OYP8G09	−	−	Environmental	This study
		OYP8G12	−	−	Environmental	This study
*V. mimicus*	3	ATCC 33653	−	−	Clinical	[60]
		ATCC 33654	−	−	Environmental	[60]
		ATCC 33655	−	−	Clinical	[60]
Other bacterial species						
*Escherichia coli*	2	CU1	−	−	Clinical	[61]
		CU2	−	−	Clinical	[62]
*Pseudomonas aeruginosa*	2	PA103	−	−	Clinical	[63]
		PA14	−	−	Clinical	[64]

**Table 2 pathogens-11-00363-t002:** Reproducibility and repeatability of qPCR assays.

	Assay for Total *V. cholerae* (*viuB*)				Assay for *V. cholerae* O1 (*rfbO1*)			
No. of CFU/Reaction	Intra-Assay Mean	%CV ^1^	Inter-Assay Mean (Cq)	%CV	Intra-Assay Mean	%CV	Inter-Assay Mean (Cq)	%CV
300,000	19.45	0.03	19.47	0.07	19.55	0.03	19.50	0.21
30,000	22.75	0.05	22.78	0.20	22.53	0.01	22.56	0.11
3000	25.94	0.02	25.92	0.07	25.94	0.01	25.93	0.07
300	29.12	0.01	29.08	0.19	29.23	0.02	29.28	0.13
30	32.57	0.02	32.56	0.08	32.63	0.01	32.62	0.03
3	35.83	0.01	35.85	0.07	35.83	0.01	35.85	0.03
1	No amplification							
0								

^1^ %CV: percent coefficient of variation.

**Table 3 pathogens-11-00363-t003:** Target genes and sequences of primers and probes used in this study.

Target Gene	Primer and Probe	Sequence (5′-3′)	Amplicon Size (bp)	References
*viuB*	Probe	56-FAM/TCATTTGGC/ZEN/CAGAGCATAAACCGGT/3IABkFQ	77	[77]
	Forward primer	TCGGTATTGTCTAACGGTAT		
	Reverse Primer	CGATTCGTGAGGGTGATA		
*rfbO1*	Probe	56-FAM/AGAAGTGTG/ZEN/TGGGCCAGGTAAAGT/3IABkFQ	113	[77]
	Forward primer	GTAAAGCAGGATGGAAACATATTC		
	Reverse primer	TGGGCTTACAAACTCAAGTAAG		

## Data Availability

Data is contained within the article or Appendix A.

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
