# Peer review of "Assay for Evaluating the Abundance of Vibrio cholerae and Its O1 Serogroup Subpopulation from Water without DNA Extraction"

_pathogens, 2022, doi:10.3390/pathogens11030363_

Round 1
Reviewer 1 Report
In this manuscript, entitled “Assay for evaluating the abundance of Vibrio cholerae and its O1 serogroup subpopulation directly from water without DNA extraction”, the authors have optimized a method for V. cholerae detection by qPCR, which can be used on-site directly on water, without the need for DNA extraction. Overall, this work deals with the development of a method that may be of great sanitary interest to control cholera outbreaks and perhaps to prevent the transmission of other waterborne infectious diseases if it were applied to detect other pathogens.
Although this is certainly a topic of great interest and the article is well written, I think that there are some aspects of the manuscript that should be revised. The main problem is that in the text, the terms “CFUs” and “genome copies” are used interchangeably, assuming that each colony-forming-unit has a single copy of the bacterial genome. Here are some examples:
Information about the limit of detection of sample:
- Line 26-27 (Abstract): ”… limit of detection is as low as 5 × 103 genome copies/L …”
- Line 138-139 (Results): “… determined a Sample Limit of Detection (SLOD) of 5 × 103 CFU/L … (TABLE S1)”
- Table S1: information is given as No of copies/mL sample
- Line 226 (Discussion): “The limit of detection for concentrated water samples was 5 × 103 CFU/L”
Information about the limit of detection:
- Line 130-131 (Results): “Analytical sensitivity was also found to be 100% based on detection of 3 CFU/reaction (TABLE 2)”
- Table 2: CFU/reaction
- Lines 136-138: “The LOD of the assay was determined as 3 copies per reaction from the standard curve constructed using serially diluted standards of the cholerae El Tor O1 N16961 reference strain (FIGURE 2)”
- Figure 2: in the axis, it is represented the CFU/reaction, but in the legend the authors use the term copies/reaction
- Line 224 (discussion): “… assay sensitivity was as low as three copies per reaction”
- Line 363- 370 (M&M): “eries of standards were prepared in which bacterial cells were present at 3 × 105 copies, 3 × 104 copies, 3 × 103 copies, 3 × 102 copies, 30 copies and 3 copies per 10 μL of filter sterilized water. … A standard curve was generated by plotting the log value of the calculated CFU per reaction against the Cq”
The manuscript assumes that the genome copy number of V. cholerae is stable regardless of environmental conditions and is 1, but there is no empirical verification that this is the case. In recent years several articles have been published on polyploidy in bacteria, in the case of V. cholerae, Paranjape & Shashidhar (FEMS Microbiology Letters, 364, 2017, fnx190) analyzed the genome copies present in this bacterium. They determined that this number can vary between 2 and 72 depending on the available nutrients, indicating that “the ploidy is highest during the early stationary phase (56-72 per cell) and lowest in the long-term starved state”. This implies that:
- If V. cholerae cells have multiple copies of their genome, it is not appropriate to assume that each CFUs corresponds to one copy of the genome. I believe this is an error that should be corrected. To use the equivalence of 1 CFU = 1 genome copy, it should be determined that the genome copy number is indeed 1 in all conditions studied. If not, the results would have to be expressed in another way.
- If the number of copies is not stable and depends on environmental conditions, it is not appropriate to compare the results obtained with laboratory cultures and those from environmental samples. Therefore, it would be convenient to validate the results obtained in environmental samples with an alternative method.
Minor points to consider:
- In lines 114-116 is stated that “environmental water samples collected from the Gabtoli area (Dhaka, Bangladesh) (FIGURE 1) and spiked with V. cholerae reference strains.”, but not only V. cholerae reference strains were tested. Were the other strains processed in a different way?
- The information in Figure 2 is not well read
Author Response
Reviewer 1
In this manuscript, entitled “Assay for evaluating the abundance of Vibrio cholerae and its O1 serogroup subpopulation directly from water without DNA extraction”, the authors have optimized a method for V. cholerae detection by qPCR, which can be used on-site directly on water, without the need for DNA extraction. Overall, this work deals with the development of a method that may be of great sanitary interest to control cholera outbreaks and perhaps to prevent the transmission of other waterborne infectious diseases if it were applied to detect other pathogens.
Although this is certainly a topic of great interest and the article is well written, I think that there are some aspects of the manuscript that should be revised. The main problem is that in the text, the terms “CFUs” and “genome copies” are used interchangeably, assuming that each colony-forming-unit has a single copy of the bacterial genome. Here are some examples:
Information about the limit of detection of sample:
- Line 26-27 (Abstract): ”… limit of detection is as low as 5 × 103 genome copies/L …”
- Line 138-139 (Results): “… determined a Sample Limit of Detection (SLOD) of 5 × 103 CFU/L … (TABLE S1)”
- Table S1: information is given as No of copies/mL sample
- Line 226 (Discussion): “The limit of detection for concentrated water samples was 5 × 103 CFU/L”
Author’s response: We thank the reviewer for recognising the significance of this work and offering valuable suggestions. We agree with the reviewer that the use of terms “CFUs” and “genome copies” should be standardised. The term “genome copies” and “no of copies” have been changed to CFU in Table S1. In abstract (line 26 – 27), we will keep the term for limit of detection as ‘genome copies’ as it is measured in concentrated biomass directly collected from the environmental water as template for qPCR.
Information about the limit of detection:
- Line 130-131 (Results): “Analytical sensitivity was also found to be 100% based on detection of 3 CFU/reaction (TABLE 2)”
- Table 2: CFU/reaction
- Lines 136-138: “The LOD of the assay was determined as 3 copies per reaction from the standard curve constructed using serially diluted standards of the cholerae El Tor O1 N16961 reference strain (FIGURE 2)”
- Figure 2: in the axis, it is represented the CFU/reaction, but in the legend the authors use the term copies/reaction
- Line 224 (discussion): “… assay sensitivity was as low as three copies per reaction”
- Line 363- 370 (M&M): “eries of standards were prepared in which bacterial cells were present at 3 × 105 copies, 3 × 104 copies, 3 × 103 copies, 3 × 102 copies, 30 copies and 3 copies per 10 μL of filter sterilized water. … A standard curve was generated by plotting the log value of the calculated CFU per reaction against the Cq”
Author’s response: The term “copies” has been changed to “CFU” at line 140, Figure 2, line 229, line 369 – 370.
The manuscript assumes that the genome copy number of V. cholerae is stable regardless of environmental conditions and is 1, but there is no empirical verification that this is the case. In recent years several articles have been published on polyploidy in bacteria, in the case of V. cholerae, Paranjape & Shashidhar (FEMS Microbiology Letters, 364, 2017, fnx190) analyzed the genome copies present in this bacterium. They determined that this number can vary between 2 and 72 depending on the available nutrients, indicating that “the ploidy is highest during the early stationary phase (56-72 per cell) and lowest in the long-term starved state”. This implies that:
- If V. cholerae cells have multiple copies of their genome, it is not appropriate to assume that each CFUs corresponds to one copy of the genome. I believe this is an error that should be corrected. To use the equivalence of 1 CFU = 1 genome copy, it should be determined that the genome copy number is indeed 1 in all conditions studied. If not, the results would have to be expressed in another way.
- If the number of copies is not stable and depends on environmental conditions, it is not appropriate to compare the results obtained with laboratory cultures and those from environmental samples. Therefore, it would be convenient to validate the results obtained in environmental samples with an alternative method.
Author’s response: We agree with the reviewer that there is no evidence for equivalency between CFU and genome copies. The method was validated in the laboratory with CFU for all the instances and V. cholerae N16961 as a reference strain containing single copy viuB gene in its genome. But for environmental water samples evaluation, we are not able to use “CFU” and instead used “genome copies” because we used concentrated biomass directly collected from the environmental water as template for qPCR. In the method developed, number of viuB gene represents the number of genome copies. By evaluating the number of genome copies present in the environmental water sample we can estimate the possible count of V. cholerae, at least in a comparative way between similar samples. Unfortunately, since V. cholerae can enter a viable but non-culturable state in the environment, we cannot accurately estimate CFU per genome copy. We agree this is a limitation of the method.
Minor points to consider:
- In lines 114-116 is stated that “environmental water samples collected from the Gabtoli area (Dhaka, Bangladesh) (FIGURE 1) and spiked with V. cholerae reference strains.”, but not only V. cholerae reference strains were tested. Were the other strains processed in a different way?
Author’s response: All forty-six bacterial isolates were tested in the same way, we have re-structured the sentence at line 119-120 to “The qPCR assay was validated by using a blind panel of filter-sterilized environmental water samples collected from the Gabtoli area (Dhaka, Bangladesh) (FIGURE 1) and spiked with V. cholerae reference strains, other vibrio species and non-vibrio species.”
- The information in Figure 2 is not well read
Author’s response: We apologise for the issue; we have now replaced the figure.
The legend of Figure 2 is modified to “Standard curves for detection and quantification of total V. cholerae and V. cholerae O1 by qPCR. Two gene markers with fluorogenic probes, A) viuB (V. cholerae specific) and B) rfbO1 (V. cholerae O1 specific) were used. Cells of reference culture (V. cholerae N16961 El Tor O1) were serially diluted 10-fold to yield concentrations ranging 3 to 3 × 106CFU per reaction (from left to right.) Fluorescence was measured in relative units. Each reaction was done in triplicate.”
Reviewer 2 Report
The manuscript entitled “Assay for evaluating the abundance of Vibrio cholerae and its O1 serogroup subpopulation directly from water without DNA extraction” is well written and the conclusions drawn are sound. However, in my opinion several figures need to be revised and some additional experiments might strengthen the arguments presented.
My specific points:
Major:
The authors mention several times throughout the manuscript that the measurements are taken directly from water samples when they are concentrated via size exclusion centrifugation. On the one hand I would be interested if this method can be used to detect Vibrio cholerae directly from water samples during an ongoing outbreak (maybe spike water samples with known concentrations that occur during outbreaks). On the other hand, I think it would be beneficial to compare this technique with established methods to show the benefits even if a centrifugation step is necessary.
Please discuss the differences between your previous study (10.3390/pathogens9121053) and the current one (different mode of concentration, different qPRC protocol). I think this can help to understand the novelty of this study.
Figure 2: The labeling is not readable due to overlapping question marks and the lines are not straight especially on the lower right end.
Figure 3: Please use the same font as in figure 4 for the labeling of the axis.
Minor:
I don’t think that showing the Google maps locations in figure 1are needed especially as you give the coordinates in the material and methods chapter.
In line 37, 38 and 48 you talk extensively about death rates of cholera. However, I think one or two sentences are necessary to explain the discrepancy of yearly cases/deaths per cases/deaths in 2021 as well as the death rate in general (2.9million cases and 95,000 deaths) compared to the Yemen outbreak (2.5 million cases and 4,000 deaths).
Table 2: I think the addition of 1 and 0 CFU/reaction might be helpful.
Table S1: Please show the number of copies per L instead of ml. This will help to compare these values as you use genome copies/L regularly.
Author Response
The manuscript entitled “Assay for evaluating the abundance of Vibrio cholerae and its O1 serogroup subpopulation directly from water without DNA extraction” is well written and the conclusions drawn are sound. However, in my opinion several figures need to be revised and some additional experiments might strengthen the arguments presented.
My specific points:
Major:
The authors mention several times throughout the manuscript that the measurements are taken directly from water samples when they are concentrated via size exclusion centrifugation. On the one hand I would be interested if this method can be used to detect Vibrio cholerae directly from water samples during an ongoing outbreak (maybe spike water samples with known concentrations that occur during outbreaks). On the other hand, I think it would be beneficial to compare this technique with established methods to show the benefits even if a centrifugation step is necessary.
Author’s response: Thank you for your suggestion, we collected samples from Gabtoli area, Dhaka, Bangladesh, an endemic area of cholera outbreak, with frequent cholera infections among people living in the surrounding areas [Nasreen, T.H., N.A.S.; Islam, M.T.; Orata, F.D.; Kirchberger, P.C.; Case, R.J.; Alam, M.; Yanow, S.K.; Boucher, Y.F. Simultaneous Quantification of Vibrio metoecus and Vibrio cholerae with Its O1 Serogroup and Toxigenic Subpopulations in Environmental Reservoirs. Pathogens 2020, 9, 1053]. From our data, the samples collected had abundance of V. cholerae between 3.7 × 104 to 3.9 × 104 genome copies/L and V. cholerae O1 was found at 4.7 × 103 to 5.4 × 103 genome copies/L. The samples we collected indicated an infectious dose to a host when water is ingested, where the infectious dose for toxigenic V. cholerae O1 is typically around 104 - 106 cells, whereas the infective dose for non-O1 strains is around 106 - 109 cells.
As the main challenge for direct quantification using existing qPCR methods is limit of detection (LOD), size-exclusion centrifugation is to concentrate the bacteria in the samples. As demonstrated in our study, the limit of detection of V. cholerae for concentrated water samples was 5 × 103 CFU/L; without the concentration procedure, the limit of detection of V. cholerae in environmental water samples would be 3 × 106 CFU/L.
Please discuss the differences between your previous study (10.3390/pathogens9121053) and the current one (different mode of concentration, different qPRC protocol). I think this can help to understand the novelty of this study.
Author’s response: Our previous study was not a direct quantification and requires a DNA extraction step prior to qPCR. Furthermore, in this study, we employed Chai Open qPCR, which is a low cost, portable qPCR thermocycler. The qPCR mix used was also a special cocktail stable at 4C for an extended period of time.
Figure 2: The labeling is not readable due to overlapping question marks and the lines are not straight especially on the lower right end.
Author’s response: We apologise for the issue; we have now replaced the figure.
Figure 3: Please use the same font as in figure 4 for the labeling of the axis.
Author’s response: Thank you for the suggestion, we have changed the font in figure 3.
Minor:
I don’t think that showing the Google maps locations in figure 1are needed especially as you give the coordinates in the material and methods chapter.
Author’s response: Thank you for your insights, the reason to show the map is for clarity, and it will be clearer to the reader the location of the sampling site in Bangladesh, to which many readers are not familiar with.
In line 37, 38 and 48 you talk extensively about death rates of cholera. However, I think one or two sentences are necessary to explain the discrepancy of yearly cases/deaths per cases/deaths in 2021 as well as the death rate in general (2.9million cases and 95,000 deaths) compared to the Yemen outbreak (2.5 million cases and 4,000 deaths).
Author’s response: Thank you for your suggestion, we have added a sentence at line 39 – 41 explaining the discrepancy.
Table 2: I think the addition of 1 and 0 CFU/reaction might be helpful.
Author’s response: There was not any amplification for 0 and 1 CFU/reaction and we have now indicated in table 2.
Table S1: Please show the number of copies per L instead of ml. This will help to compare these values as you use genome copies/L regularly.
Author’s response: As per suggestion by reviewer 1, in table S1, No of copies/mL has been changed to CFU/mL. We used CFU/mL to illustrate the true dynamic between volume of the water sample and the concentrate used in the protocol.
Reviewer 3 Report
Authors in the manuscript “Assay for evaluating the abundance of Vibrio cholerae and its O1 serogroup subpopulation directly from water without DNA extraction” describe a sensitive molecular quantification assay by qPCR, which can be used on-site in low resource settings directly on water without the need for DNA extraction. The method exhibited high specificity for total V. cholerae and V. cholerae O1 and sensitivity. The method requires concentration of biomass but the system is not complex . The ability to perform qPCR directly on water samples, and the portability of equipment speeds up the analysis. So, the work is of interest in the context of áreas areas of incidence of cholera.
Lines 37 and 38. The phrase offers contradictory data. Could authors rewrite it?
Introduction. Could authors refer to other detection methods like CARDFISH?
Girard, L., Peuchet, S., Servais, P., Henry, A., Charni-Ben-Tabassi, N., and Baudart, J. (2017) Spatiotemporal dynamics of total viable Vibrio spp. in a NW Mediterranean coastal area. Microb Environ 32:210-218.
Heidelberg, J.F., Heidelberg, K.B., and Colwell, R.R. (2002) Bacteria of the gamma-subclass Proteobacteria 2012) Rapid and sensitive quantification of Vibrio cholerae and Vibrio mimicus associated with zooplankton in Chesapeake Bay. Appl Environ Microbiol 68:5498-5507.
Schauer, S., Sommer, R., Farnleitner, A.H., and Kirschner, A.K. (icus cells in water samples by use of catalyzed reporter deposition fluorescence in situ hybridization combined with solid-phase cytometry. Appl Environ Microbiol 78:7369-7375.
Kirschner, A.K.T., Schauer, S., Steinberger, B., Wilhartitz, I., Grim, C.J., Huq, A., Colwell, R.R., Herzig, A., and Sommer, R. (2019) Interaction of Vibrio cholerae non-O1/non-O139 with copepods, cladocerans and competing bacteria in the large alkaline lake Neusiedler See, Austria. Microb Ecol 61:496-506.
Line 195. Cultured better than revived. The phrase seems to refer to resuscitation process and it is not, it refers to the loss of culturability of VBNC cells.
Lines 193-208. Excessive references to the VBNC state when in this work they do not demonstrate the possibility of detecting VBNC V. cholerae populations. Why have VBNC cultures not been used to validate the method?
Lines 305-306. How have environmental V. cholerae strains, isolated in this study, been identified?
Line 307. KCl, not KCL. Concentrations of components of Dynamite qPCR Master mix? Lack of this information prevents verification of the method.
Author Response
Reviewer 3
Authors in the manuscript “Assay for evaluating the abundance of Vibrio cholerae and its O1 serogroup subpopulation directly from water without DNA extraction” describe a sensitive molecular quantification assay by qPCR, which can be used on-site in low resource settings directly on water without the need for DNA extraction. The method exhibited high specificity for total V. cholerae and V. cholerae O1 and sensitivity. The method requires concentration of biomass but the system is not complex. The ability to perform qPCR directly on water samples, and the portability of equipment speeds up the analysis. So, the work is of interest in the context of áreas areas of incidence of cholera.
Lines 37 and 38. The phrase offers contradictory data. Could authors rewrite it?
Author’s response: Thank you for your suggestion, we have added a sentence at line 39 – 41 explaining the discrepancy.
Introduction. Could authors refer to other detection methods like CARDFISH?
- Girard, L., Peuchet, S., Servais, P., Henry, A., Charni-Ben-Tabassi, N., and Baudart, J. (2017) Spatiotemporal dynamics of total viable Vibrio in a NW Mediterranean coastal area. Microb Environ 32:210-218.
- Heidelberg, J.F., Heidelberg, K.B., and Colwell, R.R. (2002) Bacteria of the gamma-subclass Proteobacteria associated with zooplankton in Chesapeake Bay. Appl Environ Microbiol 68:5498-5507.
- Schauer, S., Sommer, R., Farnleitner, A.H., and Kirschner, A.K. (icus cells in water samples by use of catalyzed reporter deposition fluorescence in situ hybridization combined with solid-phase cytometry. Appl Environ Microbiol 78:7369-7375.
- Kirschner, A.K.T., Schauer, S., Steinberger, B., Wilhartitz, I., Grim, C.J., Huq, A., Colwell, R.R., Herzig, A., and Sommer, R. (2019) Interaction of Vibrio choleraenon-O1/non-O139 with copepods, cladocerans and competing bacteria in the large alkaline lake Neusiedler See, Austria. Microb Ecol 61:496-506.
Author’s response: Thank you for your suggestion, we have added the references into the introduction, line 98 - 100.
Line 195. Cultured better than revived. The phrase seems to refer to resuscitation process and it is not, it refers to the loss of culturability of VBNC cells.
Author’s response: We have changed the term “Revived” to “cultured”.
Lines 193-208. Excessive references to the VBNC state when in this work they do not demonstrate the possibility of detecting VBNC V. cholerae populations. Why have VBNC cultures not been used to validate the method?
Author’s response: From previous experimentation, we have not found differences in the quantification of VBNC and vegetative cells using qPCR, even when using colony PCR (cells directly added to the PCR mix) with no prior DNA extraction (unpublished results), so we did not think it necessary to demonstrate this in the manuscript. Colony PCR is used routinely for amplification of a gene marker to type V. cholerae and works well even with a small number of cells.
Lines 305-306. How have environmental V. cholerae strains, isolated in this study, been identified?
Author’s response: In this study we did not employ any culture method to isolate V. cholerae from the environmental water sample. We detected and quantified the number of total V. cholerae and V. cholerae O1 by qPCR directly from water sample. In lines 340 to 345: “previously developed species-specific primers [78] were used to detect and quantify V. cholerae in this assay. Briefly, the viuBgene encoding vibriobactin utilization protein B was used to quantify total V. cholerae (all serogroups), as it is a single copy gene present in all V. cholerae with only divergent homologs present in other species. To detect and quantify V. cholerae from the O1 serogroup, we used specific primers and probe to amplify the rfbO1 gene essential for the synthesis of this antigen (TABLE 3) [78].” We have routinely isolated strains from the same site as part of a cholera monitoring program. Strains isolated are usually typed by sequencing their viuB gene.
Line 307. KCl, not KCL. Concentrations of components of Dynamite qPCR Master mix? Lack of this information prevents verification of the method.
Author’s response: Apologies for the typing errors, we have now edited the term as “KCl” at line 352. Dynamite qPCR Master mix is a proprietary mix, developed and distributed by the Molecular Biology Service Unit (University of Alberta, Canada), further communication may be required to disclose the concentration of the components used to prepare the master mix. It is readily available for purchase there. This is similar to other proprietary mixes (from companies like IDT).
Round 2
Reviewer 1 Report
The current version of the manuscript has improved over the previous one, but I believe it still fails to answer the main questions raised by the reviewers. For the sake of brevity, I will only include the answers of the authors I would like to comment on (I agree with the rest).
Author’s response: We thank the reviewer for recognising the significance of this work and offering valuable suggestions. We agree with the reviewer that the use of terms “CFUs” and “genome copies” should be standardised. The term “genome copies” and “no of copies” have been changed to CFU in Table S1. In abstract (line 26 – 27), we will keep the term for limit of detection as ‘genome copies’ as it is measured in concentrated biomass directly collected from the environmental water as template for qPCR.
I don't understand why in the abstract "genome copies" term is maintained, however, in lines 210 or 320 the term CFU/L is used (already in the first version) to refer to what I believe is the same concept.
Author’s response: We agree with the reviewer that there is no evidence for equivalency between CFU and genome copies. The method was validated in the laboratory with CFU for all the instances and V. cholerae N16961 as a reference strain containing single copy viuB gene in its genome. But for environmental water samples evaluation, we are not able to use “CFU” and instead used “genome copies” because we used concentrated biomass directly collected from the environmental water as template for qPCR. In the method developed, number of viuB gene represents the number of genome copies. By evaluating the number of genome copies present in the environmental water sample we can estimate the possible count of V. cholerae, at least in a comparative way between similar samples. Unfortunately, since V. cholerae can enter a viable but non-culturable state in the environment, we cannot accurately estimate CFU per genome copy. We agree this is a limitation of the method.
I agree, obviously the term CFU/L cannot be used with environmental samples of which the physiological state is unknown. However, I believe that with this response it is not answered the question posed, nor even discussed the possible limitations of these assays.
Author Response
Reviewer: I don't understand why in the abstract "genome copies" term is maintained, however, in lines 210 or 320 the term CFU/L is used (already in the first version) to refer to what I believe is the same concept.
Author’s response: We have removed the use of genome copies throughout the text (see explanation below)
Reviewer: I agree, obviously the term CFU/L cannot be used with environmental samples of which the physiological state is unknown. However, I believe that with this response it is not answered the question posed, nor even discussed the possible limitations of these assays.
Author’s response: The reviewer raises a good point about using the most accurate terminology. Since the assay is calibrated using live cells in environmental water (standard curve), for which the CFU count is known (the culture used for inoculating the standard curve is plated to get an accurate CFU count), we are really measuring CFUs in the water samples. The number of genome copies in cells used for the standard curve is likely similar to that in the environmental samples, as they are in the same water under the same conditions. We let the cells sit in the filtered environment water for 30 min before processing it for the standard curve (we added that detail to the methods). Our hesitation in using the term CFU was that if one did a plate count in parallel, the results of the qPCR would not be consistent, as V. cholerae can go in VBNC quite rapidly after being in contact with room temperature water (REF: doi: 10.1038/ismej.2017.121). But upon further reflection, it is much more accurate to use CFU / L rather than genome copies. We have added text in the methods in section 4.4 to precise that when we use CFU / L, we mean CFU equivalent / L (if all cells maintained the ability to grow). We consider that doing a standard curve with live cells is more meaningful than using DNA fragments with a known number of gene copies. As the reviewer pointed out, measuring genome copies in the environment does not inform us accurately on the number of cells present. Despite our precautions, there might still be a discrepancy between genome copy numbers in cells used in the standard and those in environmental samples. We have added an explanation of this limitation in the discussion.
Reviewer 2 Report
Most of the request have been fulfilled and the manuscripts reached a quality level that merits publication. However, I suggest that you replace the term "directly" with "without DNA extraction" or delete it as this system clearly needs a concentration step and does therefore not work directly from water.Author Response
Reviewer: Most of the request have been fulfilled and the manuscripts reached a quality level that merits publication. However, I suggest that you replace the term "directly" with "without DNA extraction" or delete it as this system clearly needs a concentration step and does therefore not work directly from water.
Author’s response: We have removed any mention of “directly” from the text.